# Cat-PO: Cross-modal Adaptive Token-rewards for Preference Optimization in Truthful Multimodal LLMs

**Zhixiao Zheng[1], Zheren Fu[1], Zhiyuan Yao[1], Dongming Zhang[2], Zhendong Mao[1†]**
[1] University of Science and Technology of China
[2] State Key Laboratory of Communication Content Cognition, People's Daily Online
`{zhixiao.zheng, yaozhiyuan}@mail.ustc.edu.cn`
`{fzr, zdmao}@ustc.edu.cn, zhangdongming@people.cn`

## ABSTRACT

Multi-modal Large Language Models (MLLMs) have shown remarkable generative capabilities across multi-modal tasks, yet remain plagued by hallucinations where generated textual contents are semantically inconsistent with the input images. This work reveals that existing multi-modal preference optimization methods exhibit shortcomings at the preference data decoding stage. Specifically, different response tokens exhibit varying degrees of association with visual content, and consequently, their contributions to reducing hallucinations and generating high-quality responses differ. Nevertheless, most existing methods do not distinguish in their treatment, often handling them uniformly. To address this challenge, we propose a novel preference alignment method: Cross-modal Adaptive Token-rewarded Preference Optimization (Cat-PO). Building upon direct preference optimization, Cat-PO calculates hierarchical visual relevance rewards for each response token at global, local, and semantic levels. It then organically integrates these three rewards to construct a smooth reward mechanism and designs an innovative KL-based customized loss for rewarded tokens, thereby enabling fine-grained correction of hallucinatory outputs. Extensive experiments on various base models and evaluation benchmarks demonstrate that our Cat-PO can significantly reduce hallucinations and align with human preferences to enhance the truthfulness of MLLMs. [1]

## 1 INTRODUCTION

The success of Multimodal Large Language Models (MLLMs) marks a significant advancement in artificial intelligence research Liu et al. (2024b); Amirloo et al. (2024). By integrating visual information with Large Language Models (LLMs), MLLMs have demonstrated unprecedented capabilities in multimodal understanding, reasoning, and interaction Xiao et al. (2024); Pi et al. (2024); Zhang et al. (2024). However, MLLMs exhibit a notable hallucination problem, where the generated textual descriptions are inconsistent with the input visual content. This phenomenon includes describing non-existent objects, incorrect object attributes, or relationships Bai et al. (2024); Gunjal et al. (2024). The hallucination issue causes a disconnect between outputs and visual facts, severely degrading user experience and undermining the reliability of downstream applications, thereby limiting their deployment in real-world scenarios Liu et al. (2024a); Liang et al. (2024).

To alleviate this issue, strategies incorporating preference learning, such as Reinforcement Learning from Human Feedback (RLHF) Christiano et al. (2017), have been widely investigated as a form of fine-tuning. The core idea is to leverage preference feedback to align model outputs with desired expectations. Recently, Direct Preference Optimization (DPO) Rafailov et al. (2023) has gained prominence for achieving excellent results without a separate reward model by simplifying complex reinforcement learning steps. Existing work demonstrates that DPO, by efficiently incorporating preference data, mitigates hallucinations in MLLMs, and improves the alignment with human

---

[†]Corresponding author.
[1]`https://github.com/gavinzzx/CatPO`

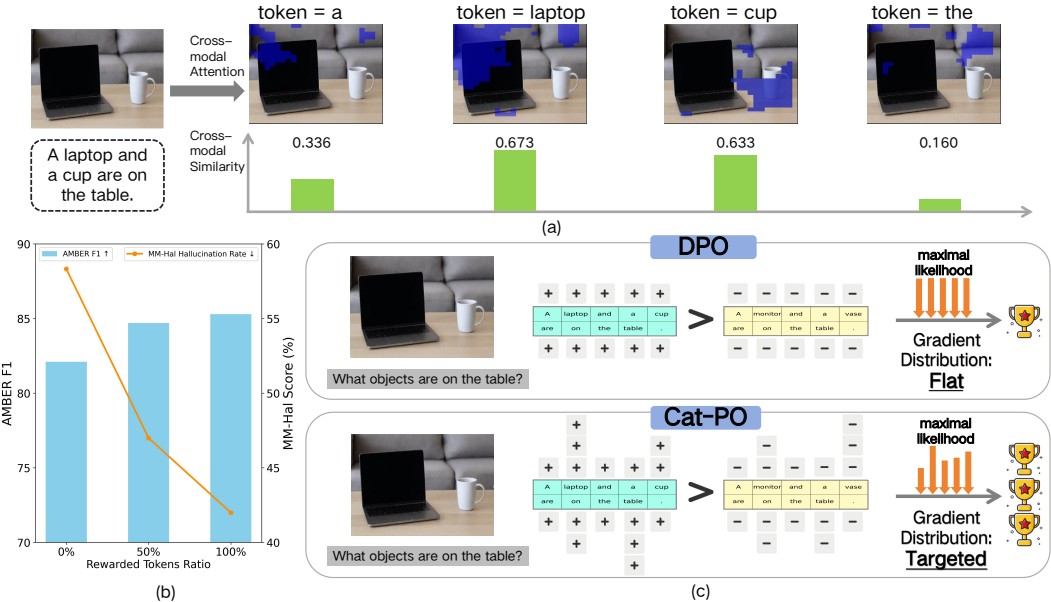

Figure 1: The motivation of our framework. (a) A visual question answering example where the model identifies "a laptop and a cup" on a table, with cross-modal attention heatmaps and cross-modal similarity scores indicating the model's visual focus and word importance in the response. (b) A performance comparison of token-rewarded DPO, showing *AMBER* F1 (↑) improving and *MM-Hal* Hallucination Rate (↓) declining as the percentage of rewarded tokens increases. (c) A comparison of standard DPO versus our Cross-modal Adaptive Token-rewarded Preference Optimization (Cat-PO). The former uses a flat gradient distribution for maximal likelihood optimization. And the latter employs a targeted gradient distribution, suggesting potentially superior performance for the latter in refining a pre-trained MLLM.

preference Li et al. (2023b); Zhu et al. (2024); Chen et al. (2024). However, in current human preference alignment processes, different response tokens processed by the model have varying degrees of relevance to the visual content, and their contributions to reducing hallucinations and generating high-quality answers also differ. As shown in Fig. 1 (a), When MLLMs process different tokens, the cross-modal attention they allocate to the image varies, and the token–image similarity also differs, indicating that different tokens exhibit distinct degrees of association with the visual content. Nevertheless, most existing works suffer from two primary limitations: (1) They overlook the varying degrees of association between different tokens in the response and the visual content, as well as their differing contributions to high-quality outputs, treating all tokens uniformly and thus lacking fine-grained correction, as depicted in the upper part of Fig. 1 (c). (2) They rely on external visual detection models, additional noise injection techniques, expensive closed-source LLM API, or even external tools, thereby neglecting the intrinsic capabilities of MLLMs and leading to a waste of existing resources and increased costs.

Therefore, how to deeply exploit token-level fine-grained alignment signals, construct a more refined DPO feedback mechanism, and fully leverage the inherent multimodal capabilities of MLLMs to reduce additional costs and overhead remains a critical issue. Motivated by this, we conducted a series of explorations. As shown by the statistical experiment in Fig. 1 (b), when we applied DPO only to the top 50% rewarded tokens in chosen responses, we observed significant improvements in hallucination metrics AMBER-F1 and MM-Hal compared to the original DPO. Furthermore, applying DPO with all rewarded tokens yielded even more outstanding results.

Building upon these explorations, we propose a Cross-modal Adaptive Token-rewarded Preference Optimization (Cat-PO). This framework fully leverages the multimodal capabilities and advantages of MLLMs to deeply mine token-level fine-grained alignment signals between text and images, using token-rewards for Cat-PO, with the aim of more effectively mitigating hallucinations. A simplified pipeline is shown in the lower part of Fig. 1 (c). Specifically, within the MLLMs, before the image

features (projected by CLIP Radford et al. (2021) and ViT) are fed into the LLM's transformer layers, we first calculate the cross-modal semantic similarity between response tokens and the image, representing the semantic relevance of tokens to visual content. Subsequently, within the transformer layers, based on the cross-modal attention of response tokens to the image, we compute the global and local relevance of each token to the visual content. Furthermore, we normalize and aggregate the three hierarchical relevance scores, map the result through an activation function, and compute the final reward for each token. Finally, we design a novel Cat-PO loss based on token-level rewards and KL divergence for further optimization. Experiments on open-source datasets and benchmarks demonstrate that our Cat-PO achieves excellent performance, significantly reducing hallucinations and improving response accuracy, thereby enhancing model truthfulness. Concurrently, this work offers a new perspective on mitigating hallucinations by fully exploiting the inherent multimodal capabilities of MLLMs without introducing external technologies or tools.

Our main contributions are summarized as follows:

1. We propose a Cross-modal Adaptive Token-rewards for Preference Optimization (Cat-PO) in MLLMs. By assigning token-rewards to highlight visually critical tokens and incorporating a fine-grained KL regularization, Cat-PO effectively reduces multimodal hallucinations.

2. We introduce a hierarchical token-rewards calculation method that relies solely on the model's inherent multimodal capabilities, without introducing any external tools or technologies. Specifically, it first computes global relevance based on cross-modal attention between text and image, then calculates local relevance based on patch entropy, and finally uses cross-modal semantic similarity for further refinement.

3. We conducted extensive experiments across multiple datasets and benchmarks to evaluate the effectiveness of Cat-PO. Notably, on the AMBER-Generation and MM-Hal benchmarks, our proposed Cat-PO outperforms existing state-of-the-art methods by 7% – 15% metrics.

## 2 RELATED WORKS

### 2.1 MLLMs HALLUCINATION

MLLMs hallucination refers to outputs that are factually inconsistent with the input image, such as identifying non-existent objects, misdescribing attributes, or misinterpreting relationships. For example, mentioning a "dog" in a landscape image that contains no animals Bai et al. (2024).

To address hallucinations in MLLMs, researchers have proposed a variety of strategies, which broadly classified as training-free or training-based Xiao et al. (2025). Training-free methods, including decoding strategies like Opera Huang et al. (2024) and VCD Leng et al. (2024). Training-based approaches reduce hallucinations through further training. Among these, preference learning such as RLHF Christiano et al. (2017) are prominent for their efficiency and effectiveness.

### 2.2 PREFERENCE LEARNING FOR HALLUCINATION

Preference learning was initially applied to LLM alignment via methods such as RLHF with PPO. These approaches typically necessitate an explicit reward model and involve complex reinforcement learning. Recently, DPO has gained widespread adoption as a simpler and more stable alternative to traditional alignment techniques. HA-DPO Zhao et al. (2023) constructs high-quality sample pairs for preference learning. POVID Zhou et al. (2024a) creates a fine-grained DPO dataset by injecting hallucinated text and adding noise to images. MDPO Wang et al. (2024) addresses "unconditional preference," where the model may ignore image. CSR Zhou et al. (2024b) iteratively constructs a preference dataset by self-generating responses, integrating visual constraints into reward modeling.

Recently, RLHF-V Yu et al. (2024) collects segment-level human preference data and performs dense DPO training. TPO Gu et al. (2024) explores token-level information in DPO for LVLMs. V-DPO Xie et al. (2024) pairs response preferences with image-contrast preferences and employs vision-guided DPO to reinforce visual context learning.

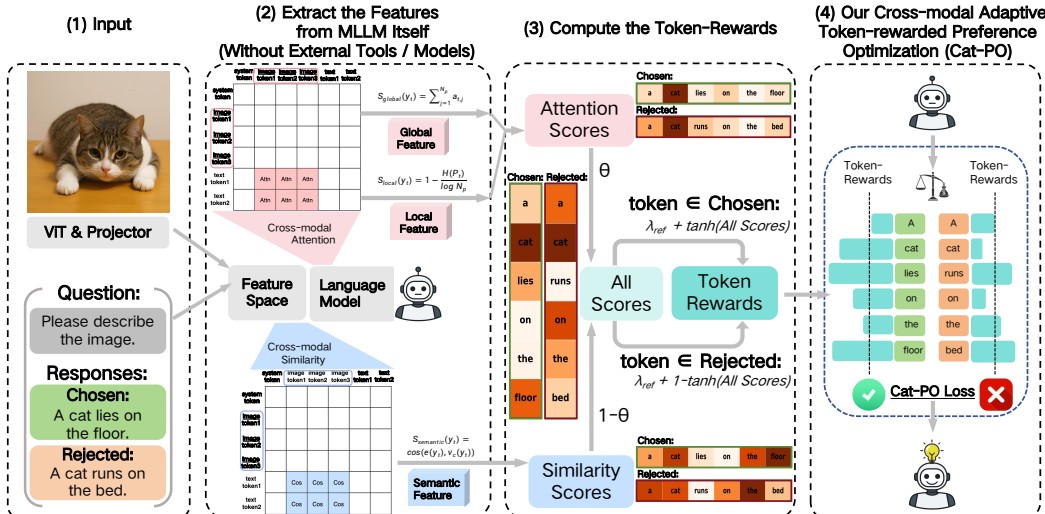

Figure 2: Overview of our proposed Cat-PO framework: (1) The visual images are first projected into the feature space via CLIP+ViT, and the textual question/response tokens are embedded by LLM tokenizer. (2) Cross-modal attention and semantic similarity are extracted in the multi-modal transformer to hierarchically form the global, local, and semantic relevance scores. (3) Token weights are computed by normalizing these scores with positive/negative sample formulas. (4) The weights are integrated into the standard DPO loss to enhance alignment and mitigate hallucinations.

## 3 METHODOLOGY

The overview of our proposed Cat-PO is illustrated in Fig. 2. We first introduce traditional DPO in Sec. 3.1. Then, we introduce the Hierarchical Visual Relevance of Tokens in Sec. 3.2 and Token-rewards in Sec. 3.3. Lastly, we describe our novel Cat-PO Loss in Sec. 3.4.

### 3.1 PRELIMINARIES: DIRECT PREFERENCE OPTIMIZATION (DPO)

DPO directly optimizes the model through a contrastive learning objective, making it more inclined to generate human-preferred responses while reducing the probability of generating dispreferred responses. DPO learns from preference data $(x, y^+, y^-) \sim \mathcal{D}$, where $x$ is the input prompt, $y^+$ is the human-preferred /chosen response, $y^-$ is the dispreferred /rejected response, and $\mathcal{D}$ is the dataset.

The DPO objective function originates from Bradley-Terry model, which assume that the human preference probability $p^*(y^+ \succ y^- \mid x)$ can be modeled via a latent reward function $r^*(x, y)$: $p^*(y^+ \succ y^- \mid x) = \sigma(r^*(x, y^+) - r^*(x, y^-))$. DPO further relates the reward function to the model's policy $\pi_\theta$ and a reference policy $\pi_{\text{ref}}$: $r_*(x, y) = \beta(\log(\pi_\theta(y \mid x)) - \log(\pi_{\text{ref}}(y \mid x)))$.

where $\beta$ is a hyperparameter controlling the ratio between reward function and policy deviation. DPO's loss can directly optimize the model to maximize the probability of generating $y^+$ and minimize generating $y^-$. For a given preference pair $(x, y^+, y^-)$, DPO loss function is defined as:

$$\mathcal{L}_{\text{DPO}} = -\log \sigma \left( \beta \left( \log \frac{\pi_\theta(y^+ \mid x)}{\pi_{ref}(y^+ \mid x)} - \log \frac{\pi_\theta(y^- \mid x)}{\pi_{ref}(y^- \mid x)} \right) \right) \tag{1}$$

By minimizing this loss function, the model $\pi_\theta$ is trained to increase the difference between the log-probabilities of preferred and dispreferred responses, relative to the reference model $\pi_{ref}$. This direct method makes DPO simpler and demonstrates comparable or superior performance to RLHF.

### 3.2 HIERARCHICAL VISUAL RELEVANCE OF TOKENS

Without any external tools or techniques, we leverage the intrinsic multimodal capabilities of MLLMs, to hierarchically compute each token's global, local, and semantic relevance to the visual input.

### 3.2.1 CROSS-MODAL ATTENTION BASED GLOBAL RELEVANCE

When MLLMs process DPO training data within the Transformer architectures, the feature representation of each token in a response interacts with visual features via a cross-modal attention mechanism. The activation distribution of these attention scores intuitively reflects the focus of specific text tokens on different image regions. Leveraging this, we define and compute a global relevance score for each token concerning the visual content, thereby quantifying its overall association with the image.

Specifically, for a given image $I$ and its corresponding preferred response $y_w$ or rejected response $y_l$ (collectively denoted $y$) from the dataset, the representation of the $t$-th token $y_t$ in $y$ serves as the query. The set of $N_p$ visual token features, $\{v_1, v_2, \ldots, v_{N_p}\}$, derived from image $I$ via a visual encoder, acts as the keys and values. The sequence of cross-modal attention scores from token $y_t$ to all $N_p$ visual tokens is denoted by $\{a_{t,1}, a_{t,2}, \ldots, a_{t,N_p}\}$.

The global relevance $S_{\text{global}}(y_t)$ is defined as the sum of the attention scores for token $y_t$:

$$S_{\text{global}}(y_t) = \sum_{j=1}^{N_p} a_{t,j} \tag{2}$$

A higher $S_{\text{global}}(y_t)$ indicates that the model attends more intensively to the entire image when processing token $y_t$, implying a stronger global alignment between the token and the visual content.

### 3.2.2 PATCH ENTROPY BASED LOCAL RELEVANCE

Although the global relevance score $S_{\text{global}}(y_t)$ captures the overall association between response tokens and visual content, it does not reveal whether attention is concentrated on key regions or dispersed across the image. Typically, focused attention indicates a strong link to specific local information, while dispersed attention suggests higher uncertainty or weaker visual grounding.

To accurately characterize the focusing pattern within this attention distribution, we leverage the concept of information entropy to compute the **patch entropy scores** for each token $y_t$ based on its image attention distribution. First, we extract the cross-modal attention vector $a_t = [a_{t,1}, a_{t,2}, \ldots, a_{t,N_p}]$ for token $y_t$ with respect to all $N_p$ image patches, where $a_{t,j}$ represents the attention strength of $y_t$ towards the $j$-th image patch. Next, we normalize the attention strengths in $a_t$ to form a probability distribution $P_t = [P_{t,1}, P_{t,2}, \ldots, P_{t,N_p}]$, and $P_{t,j} = a_{t,j} / \sum_{k=1}^{N_p} a_{t,k}$. We then compute the patch entropy $H(P_t)$ of this probability distribution. To ensure numerical stability in the logarithm, a small epsilon value $\epsilon$ (e.g., $10^{-12}$) is introduced:

$$H(P_t) = -\sum_{j=1}^{N_p} P_{t,j} \log(P_{t,j} + \epsilon) \tag{3}$$

This entropy value $H(P_t)$ measures the uncertainty or dispersion of the attention distribution. Subsequently, for $N_p > 1$, we normalize the computed entropy $H(P_t)$, and the Patch Entropy Score, $S_{\text{entropy}}(y_t)$, is then defined as 1 minus this normalized entropy:

$$S_{\text{local}}(y_t) = 1 - \frac{H(P_t)}{\log N_p} \quad (\text{for } N_p > 1) \tag{4}$$

A higher $S_{\text{local}}(y_t)$ score indicates lower entropy in the attention distribution, implying that attention is more sharply focused on a few image patches. This generally signifies a stronger degree of association between the token $y_t$ and specific local regions of the image.

### 3.2.3 CROSS-MODAL SIMILARITY BASED SEMANTIC RELEVANCE

Beyond the global and local relevance, we exploit a prior semantic signal obtained from a pretrained cross–modal encoder to quantify the semantic alignment between response tokens and visual content.

Given a sample $(I, y)$, let the embedding of the $t$-th token be $\mathbf{e}(y_t)$. The image $I$ is divided into $N_p$ patches, each encoded as a visual feature $\{\mathbf{v}_1, \ldots, \mathbf{v}_{N_p}\}$. With the cross–modal attention weights $\alpha_{t,j}$ (normalized over patches), we obtain a context–aware visual vector: $\mathbf{v}_c(y_t) = \sum_{j=1}^{N_p} \alpha_{t,j} \mathbf{v}_j$
The semantic relevance score is then defined as

$$S_{\text{semantic}}(y_t) = \cos\big(\mathbf{e}(y_t), \mathbf{v}_c(y_t)\big) = \frac{\mathbf{e}(y_t) \cdot \mathbf{v}_c(y_t)}{\|\mathbf{e}(y_t)\| \, \|\mathbf{v}_c(y_t)\|}. \tag{5}$$

This score captures the semantic relevance between the token representation and the visual content of its most attended region, complementing the attention-based global and local relevance.

### 3.3 TOKEN WEIGHTING SCHEME

**Unified Visual Relevance Score:** After obtaining hierarchical relevance scores for every response token $y_i$, we normalize all scores to $[0, 1]$ and fuse them into a unified visual relevance score.

$$s_i = \alpha\big[\, 0.5 * S_{\text{global},i} + 0.5 * S_{\text{local},i}\,\big] + (1 - \alpha)\, S_{\text{semantic},i}, \quad \alpha \in [0, 1]. \tag{6}$$

Here, $\alpha$ balances the attention branch (global & local) against the semantic branch.

**Smooth Mapping to Token Weights:** Directly injecting $s_i$ into the loss may yield unstable gradients. We therefore apply a smooth non-linearity: $T_i = \tanh(s_i) \in (0, 1)$, and introduce a base weight $\lambda_{\text{ref}} > 0$ to maintain a controlled dynamic range:

$$w_i = \begin{cases} \lambda_{\text{ref}} + T_i, & y_i \in y^+, \\ \lambda_{\text{ref}} + (1 - T_i), & y_i \in y^-. \end{cases} \tag{7}$$

This design (i) rewards tokens in the preferred response that strongly align with the image ($T_i \uparrow$), and (ii) penalises hallucinated or weakly aligned tokens in the dispreferred response (($1 - T_i) \uparrow$).

### 3.4 WEIGHTED INTEGRATION AND KL-REGULARISED CAT-PO LOSS

Incorporating token weights $\{w_t^+, w_t^-\}$ and token-level KL into the DPO loss yields the Cat-PO loss.

**Weighted DPO.** For a preference pair $(y^+, y^-)$, we weight the log-likelihood ratio of the policy $\pi_\theta$ and the reference $\pi_{ref}$. The weighted policy $\pi_\theta^{(w)}$ is defined as $\pi_\theta^{(w)} = \sum_t \big(w_t^+ \log \pi_\theta(y_t^+ \mid h_t^+) - w_t^- \log \pi_\theta(y_t^- \mid h_t^-)\big)$, and the weighted reference $\pi_{ref}^{(w)}$ is defined as $\pi_{ref}^{(w)} = \sum_t \big(w_t^+ \log \pi_{ref}(y_t^+ \mid h_t^+) - w_t^- \log \pi_{ref}(y_t^- \mid h_t^-)\big)$. Thus, the weighted DPO loss is defined as:

$$\mathcal{L}_{\text{wDPO}} = -\log \sigma\big[\beta\big(\pi_\theta^{(w)} - \pi_{ref}^{(w)}\big)\big] \tag{8}$$

**Token-weighted KL regulariser.** To keep the policy close to the reference and to stabilise training, with a regularisation strength $\lambda > 0$, we add a token-level, weight-modulated KL penalty:

$$\mathcal{L}_{\text{KL}} = \lambda\left(\sum_t w_t^+ \, \text{KL}\big[\pi_\theta(\cdot \mid h_t^+) \,\|\, \pi_{ref}(\cdot \mid h_t^+)\big] + \sum_t w_t^- \, \text{KL}\big[\pi_\theta(\cdot \mid h_t^-) \,\|\, \pi_{ref}(\cdot \mid h_t^-)\big]\right), \tag{9}$$

The final **Cat-PO Loss objective** is:

$$\mathcal{L}_{\text{Cat-PO}} = \mathcal{L}_{\text{wDPO}} + \mathcal{L}_{\text{KL}} \tag{10}$$

Minimising equation 10 enables the policy model encode human preferences and fine-grained token–vision alignments, suppressing hallucinations while preserving generation quality.

## 4 EXPERIMENTS

### 4.1 DATASETS AND METRICS

**Training Data:** Our experiments primarily employ the widely used RLHF-V dataset Yu et al. (2024). It comprises 5,733 samples, each including an image, a question, a high-quality response, and a relatively low-quality response. We use these data to compute token-weights and train our model.

**Evaluation Benchmarks:** To comprehensively evaluate the model's performance in reducing hallucinations and enhancing general capabilities, we employ several widely used benchmarks:

For hallucination evaluation, **AMBER** Wang et al. (2023) is a LLM-free benchmark which consists of two main sub-tasks: (a) *Discrimination:* Determining whether a given statement about an image

Table 1: Performance comparison on the Discrimination and Generative of AMBER Wang et al. (2023), MM-Hal Sun et al. (2023), LLaVA-Bench Liu et al. (2023b) and SEED Li et al. (2023a) benchmarks. All methods are based on LLaVA-v1.5-7B and -13B Liu et al. (2023b) models with the RLHF-V Yu et al. (2024) dataset, with the best results highlighted in **bold**.

| Method | AMBER-Disc | | AMBER-Gene | | | MM-Hal | | LLaVA ↑ | SEED ↑ |
|---|---|---|---|---|---|---|---|---|---|
| | Acc ↑ | F1 ↑ | CHAIR ↓ | Hal ↓ | Cog ↓ | Score ↑ | Rate ↓ | | |
| LLaVA-v1.5-7B | 71.7 | 74.3 | 7.8 | 36.4 | 4.2 | 2.01 | 61.4 | 65.6 | 66.1 |
| + DPO Rafailov et al. (2023) | 77.5 | 82.1 | 5.7 | 27.3 | 2.6 | 2.14 | 58.3 | 69.1 | 66.4 |
| + CSR Zhou et al. (2024b) | 73.2 | 76.1 | 5.4 | 25.5 | 2.6 | 2.05 | 60.4 | 68.9 | 65.9 |
| + POVID Zhou et al. (2024a) | 71.9 | 74.7 | 5.7 | 26.9 | 3.0 | 2.26 | 55.2 | 68.2 | 66.1 |
| + RLHF-V Yu et al. (2024) | 74.8 | 78.5 | 5.5 | 26.3 | 2.5 | 2.02 | 60.4 | 68.0 | 66.1 |
| + V-DPO Xie et al. (2024) | - | 81.6 | 5.6 | 27.3 | 2.7 | 2.16 | 56.0 | - | - |
| + TPO Gu et al. (2024) | **79.3** | 85.0 | - | - | - | 2.47 | 51.0 | 70.2 | 66.6 |
| **+ Cat-PO (Ours)** | 78.0 | **85.3** | **4.8** | **23.7** | **2.1** | **2.74** | **42.0** | **70.3** | **67.0** |
| LLaVA-v1.5-13B | 71.3 | 73.1 | 7.0 | 33.1 | 3.3 | 2.38 | 53.1 | 73.1 | 68.2 |
| + DPO (Rafailov et al., 2023) | 83.2 | 86.9 | 6.1 | 26.3 | 2.7 | 2.47 | 51.0 | 72.8 | 68.6 |
| + TPO (Gu et al., 2024) | **83.9** | **88.0** | - | - | - | 2.72 | 45.8 | 72.8 | 68.7 |
| **+ Cat-PO (Ours)** | 82.9 | **88.0** | **4.3** | **22.0** | **1.6** | **2.85** | **42.0** | **74.3** | **69.2** |

is correct or not. (b) *Generation:* Generating a descriptive text based on the image and question. **MM-Hal** Sun et al. (2023) evaluates response-level hallucination rate and informativeness. It requires GPT-4 to compare model outputs with human responses and object labels for evaluation.

For general capability evaluation, **LLaVA-Bench** Liu et al. (2023b) is a comprehensive benchmark that uses GPT-4 scoring to evaluate model generalization. **SEED-Bench** Li et al. (2023a) is a large-scale multimodal benchmark assessing visual understanding and text/image generation.

## 4.2 IMPLEMENTATION DETAILS

In our experiments, we leverage the widely adopted LLaVA-v1.5 Liu et al. (2023a) and Qwen2.5-VL Bai et al. (2025) series models to evaluate the scalability and effectiveness. The training of main experiment was performed over 6 epochs with an effective batch size of 32, implemented through gradient accumulation. And the DPO hyperparameter $\beta_{\mathrm{DPO}}$ set to 0.1.

## 4.3 MAIN RESULTS

We compare Cat-PO with advanced preference alignment methods, which include: **DPO** Rafailov et al. (2023). **CSR**, Zhou et al. (2024b) A calibrated self-rewarding method. **POVID**, Zhou et al. (2024a) A GPT-4V based alignment method. **RLHF-V**, Yu et al. (2024) A method that segments human preference collection. **TPO**, Gu et al. (2024) A DPO variant employing self-calibrated, visually anchored rewards. **V-DPO**, Xie et al. (2024) A vision-guided DPO variant.

In table 1, we evaluate existing methods and Cat-PO across multiple benchmarks. On the AMBER-Generation and MM-Hal, Cat-PO significantly improves response quality while effectively reducing hallucinations. On the AMBER-Discrimination, it achieves competitive performance, highlighting its ability to evaluate image-related statements. Furthermore, on general benchmarks such as LLaVA-Bench and SEED-Bench, Cat-PO also remains outstanding.

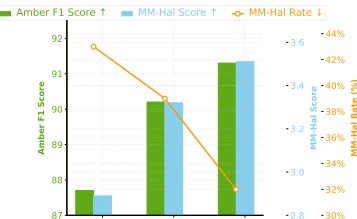

Figure 3: Performance comparison of different Qwen2.5-VL models in terms of AMBER and MM-Hal Benchmarks.

To verify the cross-model generalization of Cat-PO, we also conduct experiments on Qwen-2.5VL-3B Bai et al. (2025). As shown in Fig 3, Cat-PO achieves improvements over Qwen and Qwen+DPO on MM-Hal and AMBER benchmarks, demonstrating its strong generalization ability.

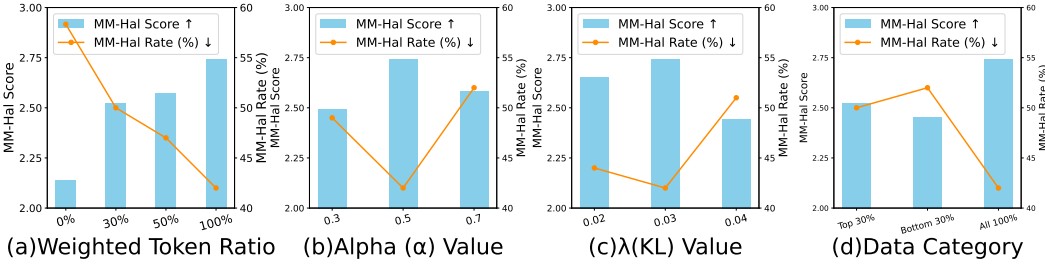

Figure 4: The performance comparison of (a) different weighting proportions, (b)(c) important hyper-parameters $\alpha$ / $\lambda_{KL}$, and (d) weighting positions in our proposed Cat-PO framework.

## 4.4 ABLATION STUDY

**Modules' Contribution.** To further validate Cat-PO's effectiveness, we conducted a comprehensive ablation study in Table 2. The results show that cross-modal attention relevance and semantic relevance play critical roles: weighting either alone improves performance, but their combination yields even greater gains. Moreover, removing the KL-based loss optimization causes a performance drop, confirming the necessity of the token-level KL term.

**Impact of Weighted-Tokens Proportion.** We further investigate how varying the proportion of weighted tokens in the chosen responses affects Cat-PO performance. As shown in Figure 4 (a), performance steadily improves with increasing weight proportion. However, applying weights to only the top 50% of tokens yields a smaller gain than to the top 30%, indicating that weighting the remaining 50% also provides a notable contribution to overall performance.

**Hyperparameter Analysis.** We examine two key hyperparameters in Cat-PO: (1) **Balance Coefficient** $\alpha$. Figure 4 (b) shows that both excessively large and small values of $\alpha$ impair performance, underscoring the need to balance cross-modal attention and semantic relevance; (2) **KL-divergence Coefficient** $\lambda_{\mathbf{KL}}$. Figure 4 (c) demonstrates that $\lambda_{\mathrm{KL}} = 0.03$

Table 2: Performance of individual Cat-PO modules.

| Modules | MM-Hal | | AMBER-Gene | |
|---|---|---|---|---|
| | Score ↑ | Rate ↓ | CHAIR ↓ | Hal ↓ |
| DPO-only | 2.14 | 58.3 | 5.7 | 27.3 |
| Attention-only | 2.34 | 55.0 | 5.3 | 25.9 |
| Similarity-only | 2.51 | 47.0 | 5.1 | 27.7 |
| Cat-PO without KL | 2.36 | 53.0 | 5.1 | 26.9 |
| Cat-PO (Ours) | **2.74** | **42.0** | **4.8** | **23.7** |

achieves the optimal trade-off between maintaining model flexibility and constraining deviation from the reference distribution.

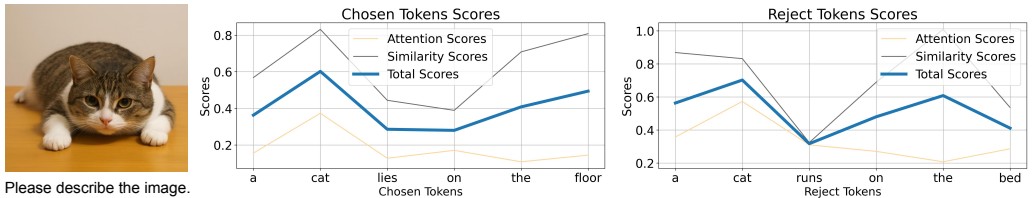

Figure 5: Token–level attention, similarity, and aggregated scores for a single example. The aggregated calculation lifts truly visual tokens (blue) while down-weighting hallucination tokens.

## 4.5 ANALYSIS EXPERIMENTS

**The Effectiveness of Token-Weights Calculation.** To further validate the effectiveness of our computed token weights, we applied weighting to either the top 30% or the bottom 30% of tokens. As shown in Figure 4 (d), weighting the top 30% of tokens yields a significant improvement compared to weighting the bottom 30%. This result confirms the accuracy of our weight computation, and demonstrates that key tokens play a decisive role in enhancing the alignment capability.

Table 3: CatPO performance on overall and adversarial subsets of POPE Li et al. (2023c).

| CatPO's Score | Acc. | Precision | F1 |
|---|---|---|---|
| Average | 85.6 | 95.2 | 84.0 |
| Adversarial (most difficult) | 84.0 (-2%) | 91.3 (-4%) | 82.5 (-2%) |

Table 4: Comparison of Cat-PO (general) and Cat-PO (with learnable fusion).

| Model | MM-Hal (↑) | Hal-Rate (↓) |
|---|---|---|
| Cat-PO (general) | 2.74 | 42% |
| Cat-PO (w/ learnable fusion) | 2.55 | 50% |

**The Analysis of Training logits.** Training logs and visualizations further demonstrate its stability. We track the evolution of the training loss and the reward margin throughout optimization. As shown in Fig. 6, the loss curve decreases smoothly, while the reward margin increases steadily without noticeable oscillations. The monitored gradient norms also remain stable. These observations indicate that Cat-PO maintains good optimization stability under token-level reward modulation.

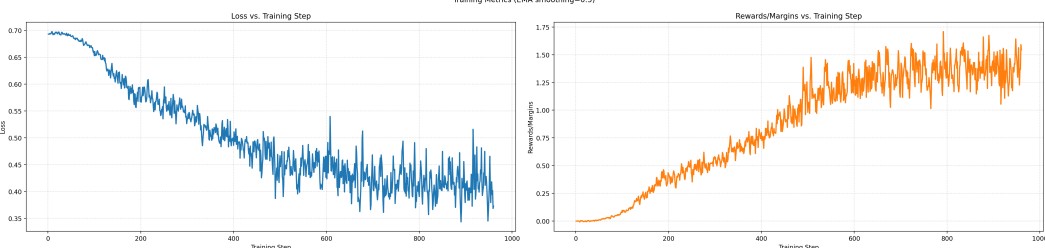

Figure 6: Training dynamics of Cat-PO, showing a smoothly decreasing loss and a steadily increasing reward margin with relatively stable gradient norms, indicating relatively stable optimization.

**Evaluation on Edge-Case Diagnostics** We additionally include POPE benchmark Li et al. (2023c) (with Object, Relation, and Adversarial subset). The Adversarial subset is explicitly designed to elicit visual hallucinations and is widely regarded as the hardest visual alignment benchmark.

Table 3 shows that Cat-PO's scores on the adversarial subset are slightly lower than average, indicating that our rewards remain relatively robust even under highly biased and adversarial edge-case settings.

**Learnable Fusion Weights.** We modified the fusion Eq. 6 to be learnable parameters as follows:

$$s_i = \gamma * S_{\text{global},i} + \delta * S_{\text{local},i} + (1 - \gamma - \delta) * S_{\text{semantic},i}, \quad \gamma, \delta \in [0,1]. \tag{11}$$

with $\gamma$ and $\delta$ are the learnable parameters. Then, we jointly optimized them within training loss. Table 4 show that introducing learnable coefficients yields the performance below the original one. Learnable fusion showed no benefit, possibly because (1) DPO does not directly supervise weight allocation, causing learnable coefficients to overfit noise, and (2) Cat-PO mainly gains from the complementary cross-modal signals, making the fixed uniform weighting a more stable design.

**Robustness Analysis of Token-reward Calculation.** To further assess robustness in edge scenarios, we visualize rare cases where attention and similarity disagree (Fig. 7). For "horse", misaligned attention is corrected by high semantic similarity, while for "train", an abnormally low similarity score is compensated by sharply focused attention. These examples show that Cat-PO's fused multi-signal scoring mutually compensates single-branch failures instead of amplifying isolated alignment errors.

From a theoretical perspective, Cat-PO computes each token-level reward by fusing three complementary cross-modal signals: global attention, local patch entropy, and cross-modal semantic similarity. A token receives a high reward only when it simultaneously exhibits strong attention strength, a stable focus pattern, and high semantic consistency with visual content. This complementarity reduces reliance on any single noisy branch and structurally limits the misalignment.

**Training Overhead Analysis.** (1) Only one-time pre-computation: Pre-computing token-level rewards for all positive and negative samples takes 2h16m18s. This cost is incurred only once and the results can be reused indefinitely. (2) We compared the training overhead of Cat-PO with DPO. Table 5 shows that Cat-PO matches DPO across all metrics, introducing only marginal overhead.

Data idx = 1520, RLHF–V Dataset      Data idx = 2290, RLHF–V Dataset

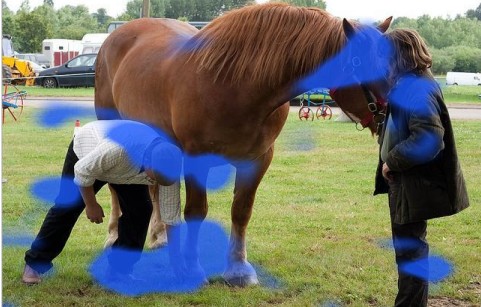 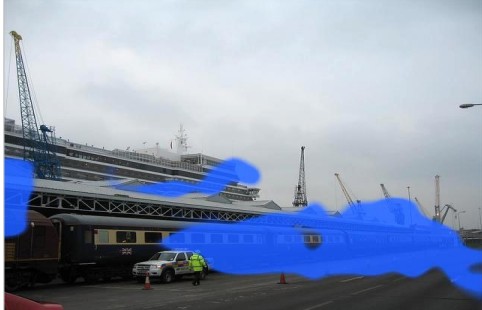

**Edge Case of Attention Misalignment, but Similarity Corrects the Rewards**

Token = "**horse**" —— Attention Map (❌)

Attention (global) = 0.08606535786179087 (❌)

Similarity (semantic) = 0.7594918817754842 (✅)

**Token–Rewards** = 0.9820886664604802 (✅)

**Edge Case of Similarity Misalignment, but Attention Corrects the Rewards**

Token = "**train**" —— Attention Map (✅)

Similarity (semantic) = 0.04862285485427614 (❌)

Attention (global) = 0.26805039819690357 (✅)

**Token–Rewards** = 0.7167500575241625 (✅)

Figure 7: Robustness Analysis in Corner Cases of Token-Reward Computation. Blue highlights indicate regions of concentrated attention. All the data are from RLHF-V Dataset Yu et al. (2024). Left Part: When visual attention for the token "horse" is misaligned (rarely distributed in horses), the high semantic similarity effectively rectifies the final reward. Right Part: Conversely, when the semantic similarity for "train" is anomalously low, the focused attention distribution (correctly distributed in train) ensures a reasonable reward. This demonstrates that the fused metrics in Cat-PO mutually compensate for single-metric misalignments, ensuring reliable preference optimization.

## 4.6 CASE STUDY

**Attention–Similarity Fusion for Token Scoring.** Figure 5 illustrates attention scores (global and local), similarity scores, and their weighted sum for a sample. Attention or similarity alone can distinguish visually critical from fact-violating tokens, validating each module's utility. However, biases exist: in the chosen sample, "floor" has a low attention score, while "on" shows low similarity. But fusing the two signals ranks critical tokens higher, enabling precise token weighting in Cat-PO and further supporting our weighting strategy.

**Comparison of Cat-PO and DPO Generations.** The comparison is presented in Appendix A.6.

Table 5: Training comparison of DPO vs. Cat-PO: average processing time of per sample and the peak memory usage.

| Model | Avg. time (s) | Peak Memory Usage (GB) |
|---|---|---|
| DPO | 2.1s | 40.420 |
| **Cat-PO** (Ours) | 2.9s (+38%) | 40.450 (+0.07%) |

## 5 CONCLUSION

In this paper, we propose Cross-modal Adaptive Token-rewarded Preference Optimization (Cat-PO) for mitigating hallucinations and improving MLLM truthfulness. Each token's global, local, and semantic relevance is computed from cross-modal attention and similarity, fused and incorporated into the DPO loss for fine-grained optimization. Experiments on public benchmarks show that Cat-PO effectively reduces hallucinations, improves response accuracy, enhances MLLMs truthfulness.

## ACKNOWLEDGMENTS

This work was supported in part by the Artificial Intelligence-National Science and Technology Major Project under Grant 2023ZD0121200.

This work was supported by the Fundamental and Interdisciplinary Disciplines Breakthrough Plan of the Ministry of Education of China (No. JYB2025XDXM103).

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

# A  Appendix

## A.1  Usage of Large Language Models in Manuscript Preparation

To ensure the clarity, fluency, and grammatical accuracy of the manuscript, large language models (LLMs) were employed during the preparation process, specifically for two core tasks: text polishing and grammatical error correction. In terms of text polishing, LLMs were used to optimize the expression of technical content (including experimental descriptions, result analyses, and discussion sections) — this involved refining sentence structure to enhance logical coherence, adjusting terminology consistency to align with academic conventions in the field of machine learning (e.g., standardizing the naming of "Cat-PO model" and "baseline DPO model" throughout the text), and improving the readability of complex statistical interpretations . For grammatical error correction, LLMs assisted in identifying and rectifying potential issues in English expression (including subject-verb agreement, tense consistency, and preposition usage) to eliminate language-related ambiguities that might affect the understanding of research findings. It is important to note that all core research content — including experimental design, data collection, model training processes, statistical analyses, and key conclusions — remained independently completed by the authors, and LLMs were only used as auxiliary tools for language optimization without altering any substantive research content.

## A.2  Limitations and Future Work.

Our approach relies solely on the intrinsic multi-modal capabilities of MLLMs, without external tools or models. This work primarily validates effectiveness and provides qualitative analysis. Future work will systematically measure resource consumption, expand evaluation metrics, and verify the resource savings from our eliminating external dependencies.

## A.3  Additional Related Works.

**In more fine-grained DPO explorations.** CHiP Fu et al. (2025) introduces visual preference optimization together with hierarchical preference optimization at the response, segment, and token levels on the text side. The token-level component is primarily based on sequence-level KL divergence derived from text probability distributions; it is not image-aware and functions only as an auxiliary term in the text-side loss. TARS Zhang et al. (2025) reformulates DPO as a min–max game. TARS introduces adaptive perturbations on vision-irrelevant tokens to induce controlled distribution shifts and combines them with a frequency-domain regularization constraint, achieving substantial improvements in visual grounding and robustness under very low data costs. AMP Zhang et al. (2024) leverages multi-level preferences to construct finer-grained preference orderings, effectively reducing the quality gap between positive and negative samples. It focuses on response-level alignment by incorporating multiple preference levels.

**In pure-text LLMs.** TGDPO Zhu et al. (2025) introduces a reward-guided DPO framework that decomposes sequence-level PPO into token-level subproblems and theoretically proves the independence of the partition function. This enables the integration of fine-grained token-level rewards into the DPO objective, improving instruction-following performance and training stability. TDPO Zeng et al. (2024) further decomposes sentence-level rewards into token-level rewards via the Bellman equation and introduces a Sequential Forward KL constraint. Through token-level optimization, TDPO enhances alignment performance in text-only generation.

**Other related works.** Hallucinations in MLLMs from several factors: training data deficiencies Li et al. (2023c); module-specific issues within MLLMs' separately trained components Guan et al. (2024); suboptimal training paradigms Ben-Kish et al. (2023); and inference-stage defects.

## A.4  Token-Level Log-Probabilities on Chosen Responses: Cat-PO vs. DPO.

To illustrate how Cat-PO affects token-level likelihood on chosen responses, Figure 8 presents a token-level case study. Cat-PO (blue curve) assigns higher log-probabilities to most tokens in chosen responses compared to DPO (green curve), as shown by the consistently positive differences (yellow bars). This suggests that Cat-PO tends to place higher likelihood on tokens in preferred responses.

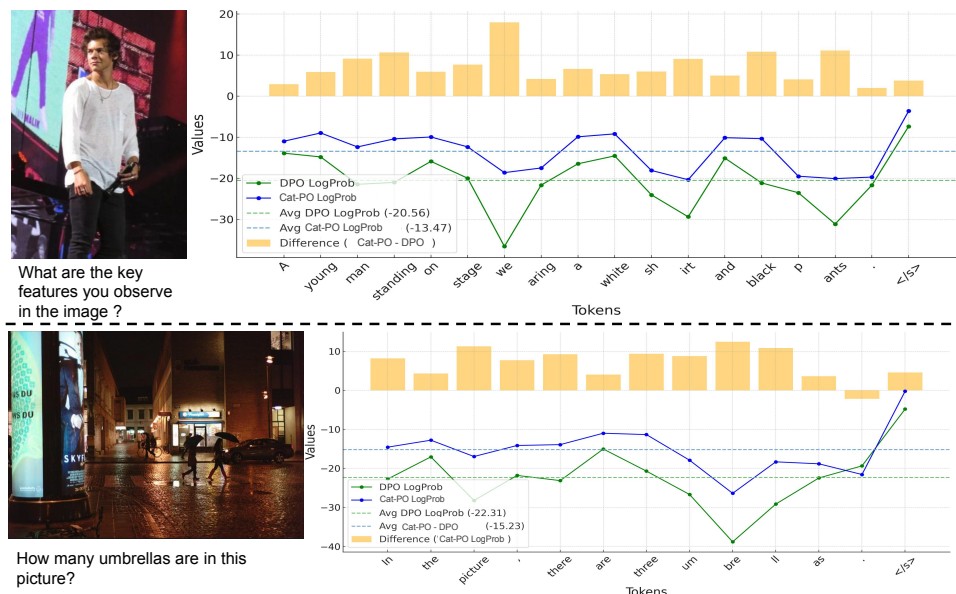

Figure 8: Log-Probability (log-prob) comparison between Cat-PO and DPO on chosen responses. The blue (Cat-PO) and green curve (DPO) represent the log-prob. The yellow bars represent differences, computed as Cat-PO minus DPO. This suggests that Cat-PO assigns higher likelihood to chosen responses than DPO.

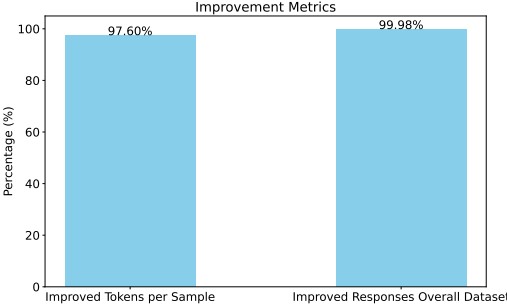

Figure 9: Improvement metrics for the model trained with Cat-PO on chosen responses. The left bar indicates the percentage of tokens within each chosen response that experienced a positive increase in their log-probability. The right bar shows the percentage of samples in the dataset that demonstrated a net growth in total log-probability under the Cat-PO model.

### A.5 Auxiliary Diagnostic: Log-Probabilities on Chosen Responses

Our supplementary statistics further examine how Cat-PO changes log-probabilities on chosen responses relative to the baseline DPO model. As shown in Figure 9, under Cat-PO, a large proportion of tokens in chosen responses exhibit increased log-probabilities, and most chosen-response samples show a positive increase in total log-probability. These results are consistent with the design of Cat-PO, which applies token-level visual relevance weighting to strengthen preferred responses under preference optimization.

### A.6 Comparison of generation in Cat-PO and DPO.

Our comparative analysis (in Figure 10) illustrates the performance differences between Cat-PO and DPO across four distinct examples. The figure contains four sub-examples demonstrating Cat-PO's consistent advantage in generating more detailed and accurate descriptions, particularly in identifying subject counts, positional relationships, and contextual details.

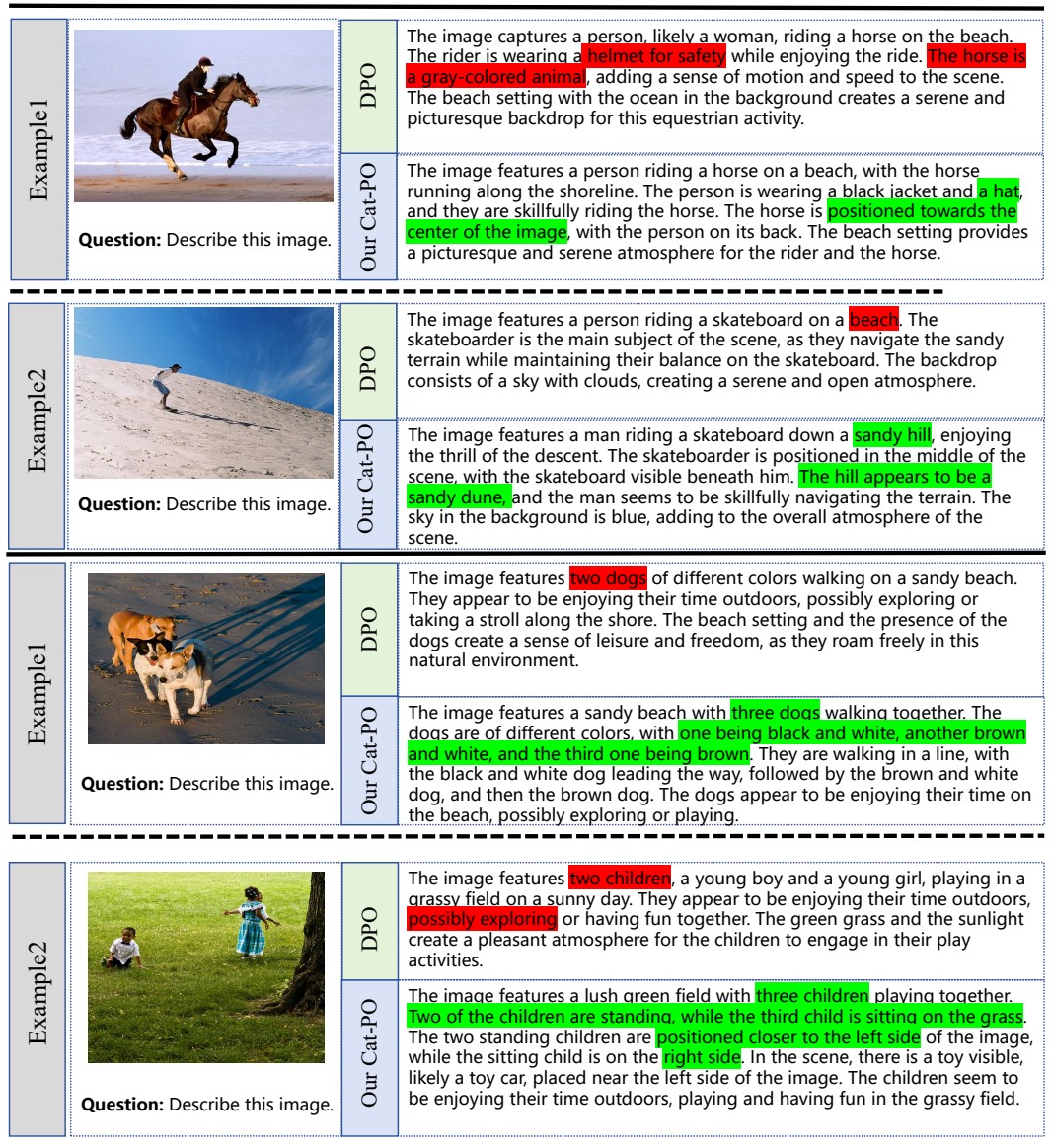

Figure 10: Four comparative examples showing generation differences between DPO and our Cat-PO.
(1) Beach horse riding: Cat-PO provides specific details about rider attire and horse movement. (2)
Sand skateboarding: Cat-PO adds contextual information about terrain and activity. (3) Beach dogs:
Cat-PO correctly identifies three dogs with distinct color patterns. (4) Children playing: Cat-PO notes
precise subject count, positions, and presence of a toy.

