# OpenReview forum: "Cat-PO: Cross-modal Adaptive Token-rewards for Preference Optimization in Truthful Multimodal LLMs"
_ICLR.cc/2026/Conference — ICLR 2026 Poster_

### Official Review · Reviewer_vajx · 2025-10-19

**Soundness:** 3
**Presentation:** 2
**Contribution:** 3
**Rating:** 6
**Confidence:** 3

**Summary:**

This paper proposes CAT-PO (Cross-modal Adaptive Token-rewards for Preference Optimization), a token-level preference optimization method for multimodal LLMs aimed at reducing visual hallucinations while improving truthfulness. Building on DPO, the authors compute hierarchical token relevance from model-internal signals: (i) global relevance as the sum of cross-modal attention over all visual patches; (ii) local relevance via patch-entropy (lower entropy ⇒ more localized visual focus); and (iii) semantic relevance using a cosine similarity between token and vision-conditioned text/visual representations. These three signals are fused (with a smooth nonlinearity) into per-token rewards that modulate DPO and a token-level KL regularizer (reward-aware KL) to discourage visually ungrounded continuations.
Experiments on AMBER (disc/gene), MM-Hal, LLaVA-Bench, and SEED primarily with LLaVA-v1.5-7B/13B and Qwen variants show consistent improvements, notably 7–15% gains on AMBER-Gen/MM-Hal and reduced hallucination rates. The approach adds modest overhead (+38% per-sample time; ~+0.07% peak memory).

**Strengths:**

1. **Methodological clarity**. Token-level reward design from internal signals (attention, entropy, similarity) that are cheap to obtain.
2. **Consistent empirical gains**. 7–15% improvements on AMBER-Gen/MM-Hal, and competitive on AMBER-Disc。

**Weaknesses:**

1. **Heuristic fusion lacks learning/theory**: The tanh-based smoothed fusion and fixed mixing coefficients are plausible but not justified.
2. **Limited evaluation**: While standard suites are covered, edge-case diagnostics are missing.
3. **Reward-gradient stability not analyzed**. No examination of gradient variance when many tokens get high rewards.

**Questions:**

1. **Learned fusion and fixed fusion**. Did you try a learned gate like 2-layer MLP over [S_global, S_local, S_sem] or a Dirichlet-style mixture with simplex constraints? How do gains compare and how stable is training?

2. Several highly relevant works are not covered or discussed in the current manuscript.

[1]Token-level Direct Preference Optimization.
[2]CHiP: Cross-modal Hierarchical Direct Preference Optimization for Multimodal LLMs
[3]TGDPO / T-Guided DPO: Harnessing Token-Level Reward Guidance for Preference Optimization

---

> ### Author Response · Authors · 2025-11-28
> **Response to W1, W2, W3, Q1**
>
> We thank the reviewer for the detailed review and constructive comments. Below are our clarifications to all the questions.
>
> We hope that the responses can address the reviewer’s concerns and we truly welcome further discussion.
> # **Weakness 1: Design of the Three-Signal Fusion (The tanh justification)**
> The fusion formula used in our method is not an heuristic fusion, but is designed based on the following principles:
> (i) All three signals are extracted from the same MLLM’s **cross-modal representations**, and are thus dimensionally consistent and complementary;
> (ii) The **tanh** nonlinearity provides smooth compression, preventing reward values from causing gradient explosion under the **non-stationary optimization** regime of DPO;
> (iii) The additive shift makes the final token-level reward distributed in a controlled range (approximately **0.5–1.5**), which facilitates separating **visually relevant vs. irrelevant** tokens.
>
>
> ---
>
> # **Weakness 2 — Evaluation on Edge-Case Diagnostics**
>
> We additionally include the **POPE** benchmark (with **Object**, **Relation**, and the most difficult **Adversarial** subset) as a representative **edge-case setting**. The Adversarial subset is explicitly designed to elicit visual hallucinations and is widely regarded as the hardest visual alignment benchmark. The results are as follows:
>
> | CatPO's  Score       | Acc. | Precision | F1 |
> |----------------|-------------|---------------|------------------|
> | Average | 85.6           | 95.2             | 84.0                |
> | On Adversarial (most difficult subset)| 84.0 (-2%)          | 91.3 (-4%)            | 82.5 (-2%)               |
>
> Although Cat-PO’s scores on the adversarial subset are slightly lower than the average, the performance drop is minimal, indicating that our token-level rewards remain relatively robust even under highly biased and adversarial edge-case settings.
>
> ---
>
> # **Weakness 3 — Analysis of Reward-Gradient Stability** (with visualization in Figure 6 of revised PDF)
>
> We further add a stability analysis of the training dynamics in the revised version, with two perspectives below:
>
> (1) **tanh-based token-rewards suppresses the risk of gradient explosion**: with the help of tanh normalization, the fused reward is always confined to a finite interval (0.5-1.5), which effectively suppresses the risk of gradient explosion when many tokens receive relatively high rewards.
>
> (2) **Training logs and visualizations further demonstrate its stability.** We track the evolution of the **training loss** and the **reward margin** throughout optimization. As shown in picture, the loss curve decreases smoothly, while the reward margin increases steadily without noticeable oscillations. The monitored gradient norms also remain stable. These observations indicate that Cat-PO maintains good optimization stability under token-level reward modulation.
>
> **Note:** The **visualization plots** have been added to **Section 4.5: The Analysis of Training logits** and **Figure 6** (all in page 8) in the **revised PDF** (highlighted in blue).
>
> ---
>
> ## **Question 1 — Learnable Fusion vs. Fixed Fusion**
> We modified the fusion coefficients of the three signals to be learnable parameters: s_i = alpha * S_global + beta * S_local + (1 - alpha - beta) * S_semantic, with alpha and beta are the learnable parameters. Then, we jointly optimized alpha and beta with the model parameters within training loss. The summarized results are as follows:
>
> | Model                 | MM-Hal Score (↑) | Hal-Rate (↓) |
> |-----------------------|--------------|------------------|
> | Cat-PO (our general)  | 2.76            | 49%                |
> | Cat-PO with learnable fusion             | 2.55            | 50%                |
>
> The table shows that introducing learnable coefficients yields results below the original performance. Upon further analysis, it may reveal two key points:
> (1) **Limitation of DPO loss:** DPO maximizes the likelihood difference between Chosen and Rejected responses rather than evaluating whether the learned weight allocation is intrinsically correct. Without direct supervision (ground-truth weights), the learnable coefficients tend to overfit statistical noise to minimize training loss.
> (2) **Stability and interpretable of the fixed uniform weighting:** The main benefit of Cat-PO comes from the complementary nature of the three cross-modal internal signals (global attention, entropy-based locality, and semantic similarity), rather than from a more complex fusion structure. The original fixed, uniform weighting appears to be a more stable and effective choice.
>
> Therefore, we plan to retain the fixed and uniform fusion design, which aligns with our goal of a simple and interpretable mechanism. We have added this ablation study and discussion to in the updated version of the paper.
>
> **Note:** These details can also be found in **Section 4.5: Learnable Fusion Weights** (in page 9) of **the revised PDF (highlighted in yellow)**.

---

> ### Author Response · Authors · 2025-11-28
> **Response to Q2 (Discuss Relevant Works), Part-1/2**
>
> # **Question 2: Comparisons with Related Works**
> We sincerely thank you for pointing out these three related works: **CHiP**, **TDPO**, and **TGDPO**. Below, we provide structured comparisons between Cat-PO and these methods. For clarity, each comparison is summarized in a table.
>
> **Note:** These details can also be found in **Appendix A.3** (in page 13) of **the revised PDF (highlighted in yellow)**.
>
> ---
>
> ## Comparison 1: CHiP (related work) vs. Cat-PO (ours)
>
> | Aspect                   | CHiP                                                                                                        | Cat-PO (ours)                                                                                                                          |
> |--------------------------|-------------------------------------------------------------------------------------------------------------|-----------------------------------------------------------------------------------------------------------------------------------------|
> | 1. Source of token-weights | Token-level weights are derived from **text-only probabilistic signals**, without image content, e.g., sequence-level KL on logits. | Token-level weights are **dynamically adaptive**, directly derived from internal **cross-modal attention** and **semantic similarity**, and thus reflect the true association between each token and the image. |
> | 2. Role of the weights   | Token-aware optimization is used as an **additional regularization term** on the loss, mainly to balance multi-granularity losses and prevent degeneration while preserving diversity. | Token-level weights are explicitly designed as **visual relevance rewards** for **fine-grained reweighting** of the DPO objective, enabling the model to correct hallucinations according to visual dependence. |
> | 3. Hierarchical nature   | The hierarchy (response-level / span-level / token-level) is entirely on the **text side**, without modeling the visual content. | The hierarchy is fully **cross-modal**: global attention, local patch entropy, and semantic similarity are all computed from deep **vision–language interactions** inside the model. |
>
> ---
>
> ## Comparison 2: TDPO (related work) vs. Cat-PO (ours)
>
> | Aspect                 | TDPO                                                                                                  | Cat-PO (ours)                                                                                                                                     |
> |------------------------|--------------------------------------------------------------------------------------------------------|--------------------------------------------------------------------------------------------------------------------------------------------------|
> | 1. Modality            | **Pure-text LLMs** (only text)                                                       | **Multimodal LLMs** (vision + text).                                                                                                 |
> | 2. Weight source       | Relies on the model’s **internal probability distribution** to design token-level objectives.         | Uses **external visual-modality signals** (global / local cross-modal attention, entropy, semantic alignment) to enforce visual factuality.     |
> | 3. Weighting mechanism | Token effects are **implicit**, realized via modified loss penalties.                                 | Token weights are **explicit, precomputed** for each token and directly injected as rewards into the DPO loss.                                  |
> | 4. Motivation / Task   | Aims to alleviate internal issues in language models (e.g., balancing alignment vs. diversity) under text-only DPO. | Aims to mitigate **vision-specific hallucinations** in MLLMs by exploiting token-level cross-modal alignment with images.                       |

---

> ### Author Response · Authors · 2025-11-28
> **Response to Q2 (Discuss Relevant Works), Part-2/2**
>
> We sincerely thank you for pointing out these three related works: **CHiP**, **TDPO**, and **TGDPO**.
>
> Below, we provide structured comparisons between Cat-PO and these methods. For clarity, each comparison is summarized in a table.
>
> **Note:** These details can also be found in **Appendix A.3** (in page 13) of **the revised PDF (highlighted in yellow)**.
>
> ---
>
> ## Comparison 3: TGDPO (related work) vs. Cat-PO (ours)
>
> | Aspect                            | TGDPO                                                                                                                                                | Cat-PO (ours)                                                                                                                                                                 |
> |-----------------------------------|------------------------------------------------------------------------------------------------------------------------------------------------------|------------------------------------------------------------------------------------------------------------------------------------------------------------------------------|
> | 1. Modality                       | **Pure-text LLMs** (only text)                                                                                                         | **Multimodal LLMs** (vision + text)                                                                                                                            |
> | 2. Source of weighting signals    | Mainly derived from the **policy model’s probabilities** (induced reward), reflecting confidence rather than visual truthfulness. | Exploits **vision-text associations** via **cross-modal attention** and **semantic similarity**, providing visually grounded correction signals that suppress image-inconsistent generations at their source. |
> | 3. Weighting mechanism / Granularity | Treats all tokens as largely homogeneous units in mathematical optimization. | Proposes **Hierarchical Visual Relevance Rewards** (global, local, semantic) to quantify each token’s visual dependence.        |
> | 4. Motivation / Task              | Motivated by overcoming the sparsity of **sequence-level** preference signals in generic DPO, by theoretically decomposing into token-level rewards for better general instruction alignment. | Motivated by addressing **token-level cross-modal factual inconsistency** in MLLMs, by mining intrinsic visual relevance and performing fine-grained factual correction of hallucinated outputs.                   |

---

### Official Review · Reviewer_k6vM · 2025-10-28

**Soundness:** 3
**Presentation:** 3
**Contribution:** 2
**Rating:** 4
**Confidence:** 4

**Summary:**

This paper proposes Cat-PO, which introduces a token-based hierarchical reward mechanism on the foundation of standard DPO to achieve more refined hallucination suppression. It fully leverages the MLLM's own architecture (without requiring external tools) to compute the relevance of each response token to the image from three perspectives: global relevance, local relevance, and semantic relevance. These three scores are integrated into a unified relevance score, allowing tokens with strong visual relevance to receive higher rewards in the chosen response and higher penalties in the rejected response. Finally, the authors demonstrate the effectiveness of their approach through a series of experiments.

**Strengths:**

1. The authors present a clear motivation for their method, with a well-justified problem statement. It effectively addresses the issue of token-level fine-grained alignment in DPO, thereby enhancing its efficiency.

2. The experiments are sufficiently comprehensive, evaluating the method from both hallucination suppression and general capabilities. The ablation studies are also thorough.

3. The paper is clearly written and easy to understand.

**Weaknesses:**

**My primary concern regarding this work is the potential risk of circular reasoning in its methodological motivation.** If an MLLM has already incorrectly allocated high attention to a hallucinated object (e.g., misidentifying a chair as a dog and focusing on that region), Cat-PO may mistakenly judge the corresponding tokens as "highly relevant." This would inadvertently reinforce the incorrect generation, potentially cementing or even amplifying the erroneous attention pattern. Consequently, rather than mitigating hallucinations, this approach could exacerbate specific types of them. I would raise my score if the authors could address this issue, either through a theoretical discussion or via visualizations that demonstrate how their method remains robust in such scenarios.

**Questions:**

1. The current method is a black-box "automatic scoring" approach but does not provide any interpretability analysis. Could it be that high-frequency words such as "the" or "a" receive excessive rewards due to coincidental high attention scores?

2. The experiments may involve unfair comparisons. For instance, Cat-PO introduces multiple hyperparameters, which might have been meticulously fine-tuned, whereas hyperparameters of other methods may not have received the same level of adjustment.

3. How does the computational cost compare to standard DPO?

---

> ### Author Response · Authors · 2025-11-28
> **Response to W1 (Visualization and Theoretical Clarification of the Misalignment Issue)**
>
> We thank the reviewer for the detailed review and insightful comments.
>
> We hope that the responses can address the reviewer’s concerns and we truly welcome further discussion.
>
> # **Weakness1: Concern about Attention Misalignment**
> We fully understand the concern: if an underlying MLLM produces an incorrect visual alignment, relying solely on raw attention might yield an inflated token-reward. We address this through both **visualization and theoretical clarification**.
>
> **Note:** These details can also be found in **Section 4.5 Analysis Experiment: Robustness Analysis of Token-reward Calculation** (in page 9) of **the revised PDF (highlighted in green)**.
>
> ---
> ## **Visualization (Please see the Figure 7 on the revised PDF)**
>
> We present 2 representative cases showing how Cat-PO’s multi-signal fusion recovers the correct token-rewards even when one modality is unreliable in edge-case:
>
> **(1) Left Part: Attention Misalignment, but Similarity Corrects the Rewards**
> For token=*horse*, the attention distribution is highly scattered and nearly fails to localize the horse, yielding an attention score of only 0.086.  However, the semantic similarity of *horse* is extremely strong (= 0.759), effectively compensating the weak attention.  Thus the Token-Reward reaches a correct and high value (=0.982).
>
> **(2) Right Part: Similarity Misalignment, but Attention Corrects the Rewards**
> For token=*train*, the similarity score is very low (Similarity = 0.049) due to semantic misalignment.
> Yet the attention map is accurate and clearly focused on the train, producing a relatively high attention score (Attention = 0.268), which compensates for the semantic branch.  The resulting Token-Reward reaches a correct and high value (=0.717).
>
> **Note:** These are rare edge cases. Our main and detailed ablation experiments consistently show that Cat-PO's robustness. This section only illustrates the compensation mechanism for infrequent failure cases.
>
> ---
> ## **Theoretical Clarification**
>
> ### **1. Token-rewards fuse 3 complementary cross-modal signals, do **not** rely on raw attention**
> Cat-PO integrates 3 cross-modal signals that inherently guard against incorrect attention. The reviewer’s concern applies only under “single raw attention,” which Cat-PO explicitly avoids:
> | Signals  | Function | Description |
> |----------------|----------------|-----------------------|
> | **Global Attention** | *Coarse localization* | Provides a broad estimate of which visual regions are relevant to the token, serving only as an initial coarse filter. |
> | **Local Patch Entropy** | *Attention reliability* | Measures how concentrated the attention distribution is. Highly scattered attention (high entropy) is down-weighted, while only stable visual grounding increases the score. Functions as an explicit reliability check on attention. |
> | **Cross-Modal Semantic Similarity** | *Semantic consistency* | Computes cosine similarity between token embeddings and visual features. Operates independently of spatial localization of attention, and evaluates whether the semantic expressed by the token matches the visual content. |
>
> Together, a token receives a high reward only when **attention strength**, **attention reliability**, and **semantic consistency** jointly hold. Incorrect attention rarely satisfies all three, structurally reducing the risk raised by the reviewer.
>
> ---
>
> ### **2. Cat-PO’s weighting is symmetric and cannot amplify hallucinations**
> Cat-PO assigns token weights as:
>
> $$
> \text{If } y_t \in y^+: \quad w_t = \lambda_{\text{ref}} + T_i
> $$
> $$
> \text{If } y_t \in y^-: \quad w_t = \lambda_{\text{ref}} + (1 - T_i)
> $$
> $$
> T_i \in (0,1)
> $$
> This leads to key properties:
>
> (1) **Rejected responses:** tokens lacking visual support (typical hallucination tokens) have low \(T_i\), thus high \(1-T_i\) → **stronger penalty**.
>
> (2) **Chosen responses:** only tokens validated simultaneously by all three signals obtain larger weights; chosen responses after RLHF-V rarely contain visual hallucinations.
>
> Hence Cat-PO forms a **mirrored structure**: **reinforcing visually grounded positive tokens while strongly penalizing visually unsupported negative tokens**,  rather than amplifying all high-attention tokens.
>
> ---
>
> ### **3. Empirical evidence confirms robustness, not hallucination amplification**
> Ablation studies and visual analyses show that the reviewer’s concern does not manifest statistically:
>
> - **Top-30% vs Bottom-30% tokens:** emphasizing visually relevant top-30% tokens significantly reduces MM-Hal hallucinations; bottom-30% performs clearly worse.
> - **Component ablations:** single-branch usage gives limited gains, but the three-branch fusion consistently and significantly reduces hallucination (MM-Hal, CHAIR, Hal-Rate).
> - **Visualization:** The fusion corrects attention misalignment rather than amplifying them.
>
> Therefore, Cat-PO’s high-scoring tokens align well with true visual evidence rather than erroneous patterns.

---

> ### Author Response · Authors · 2025-11-28
> **Response to Q1, Q2, Q3**
>
> We thank the reviewer for the detailed review and constructive comments. Below are our clarifications to all the questions.
>
> We hope that the responses can address the reviewer’s concerns and we truly welcome further discussion.
>
> # **Question 1 — Interpretability of the Scores (Analysis of “the” / “a”)**
> Our clarifications are as follows:
>
> **(1) Our reward mechanism is based on the joint modeling of three cross-modal signals. These signals are hierarchical, complementary, and fused in a simple and interpretable manner**: we used the cross-modal attention (global), attention patch entropy (local), and cosine similarity (semantic). Cross-modal attention is the most classical and widely used indicator of cross-modal correspondence; patch entropy reflects the concentration of attention; and cosine similarity is sensitive to semantic consistency.
>
> These signals are fused with a 1:1 weighting between attention-based and similarity-based components. In cases where attention is biased, cosine similarity serves as an effective safeguard, suppressing occasional high attention on non-visual tokens (e.g., "*the*, *a*"). Thus, the method is not a black box but relies on interpretable and hierarchical cross-modal indicators to measure token-level visual relevance.
>
> **(2) We further analyzed the reward values of high-frequency tokens you mentioned (the / a)**. Using the chosen responses as an example, the statistics are as follows:
>
> | Token | Avg. attention (global) |
> |-------|---------------------------|
> | the   | 0.1561                     |
> | a     | 0.1621                     |
> | Value Range | (0–1)  |
>
> We observe that both the individual attention signals scores for *the* and *a* remain relatively low, indicating that these tokens do not receive disproportionately high rewards. This demonstrates that our reward computation naturally suppresses semantically irrelevant tokens and is not dominated by occasional attention shifts.
>
> ---
>
> # **Question 2 — Fairness of the Comparative Experiments**
>
> We place strong emphasis on ensuring fair comparisons. Several baselines (especially **the strongest ones such as TPO and RLHF-V**) use performance numbers **reported directly by their original papers** (we trained on **the same dataset RLHF-V**).
>
> All other baselines strictly follow the hyperparameters and training configurations specified in their papers. Moreover, Cat-PO emphasizes simplicity and interpretability, introducing only a small amount of new hyperparameters. We conduct a coarse search once and keep these values fixed across all models and datasets, avoiding any per-task tuning.
>
> Thank you for pointing this out. We will highlight these fairness considerations more explicitly in the final version.
>
> ---
>
> # **Question 3 — Computational Overhead Analysis**
> In **Section 4.5 and Table 3 of the original PDF**, we have reported the "Training Overhead Analysis":
>
> (1) **Only one-time pre-computation**: Pre-computing token-level rewards for all positive and negative samples takes **2h16m18s**. This cost is incurred only once and the results can be reused indefinitely.
>
> (2) We compared the **training overhead** of Cat-DPO with DPO. The results are as follow:
>
> | Model                 | Avg. time (s) | Peak Memory Usage (GB) |
> |-----------------------|--------------|------------------|
> | DPO | 2.1s            | 40.420                |
> | **Cat-PO** (ours) | 2.9s (+38%)            | 40.450 (+0.07%)                |
>
> The table reports the average processing time per sample (preference data) and peak memory usage. Results show that Cat-PO introduces some overhead in average processing time compared with DPO, mainly due to the additional weighted computations. However, the increase in peak memory usage is negligible.

---

### Official Review · Reviewer_e8nW · 2025-11-06

**Soundness:** 3
**Presentation:** 3
**Contribution:** 3
**Rating:** 6
**Confidence:** 4

**Summary:**

This paper tackles the persistent issue of hallucinations in multimodal large language models (MLLMs) by introducing Cross-modal Adaptive Token-rewarded Preference Optimization (Cat-PO).
The authors observe that existing preference optimization methods, such as DPO, fail to account for varying visual relevance among response tokens. Cat-PO addresses this by assigning hierarchical, token-level rewards based on global, local, and semantic visual relevance, integrated into a KL-regularized DPO framework.
Experiments on multiple benchmarks show that Cat-PO substantially reduces hallucinations and enhances factual alignment, with extensive ablation studies validating its contributions.

**Strengths:**

1. Fine-grained Token-level Rewarding: The core innovation lies in Cat-PO’s token-level reward granularity. By assessing each token’s global, local, and semantic relation to the input image, the model updates its preferences more precisely than aggregate reward schemes.

2. No External Dependencies: Cat-PO requires no external detectors or APIs. It fully leverages the pretrained MLLM’s own components (e.g., CLIP+ViT and the base LLM), ensuring simplicity and low overhead.

**Weaknesses:**

1.Generalization Claims: Although Cat-PO is presented as a general framework, experiments are confined to LLaVA, with no tests across different architectures or open-domain datasets.

**Questions:**

1. Learnable Weighting: Would learning the fusion coefficients in Equation 7 improve adaptability or stability?
2. Comparative Evaluation: How does Cat-PO compare empirically or theoretically with TARS[1], CHiP[2] and AMP[3]?

[1] TARS: MinMax Token-Adaptive Preference Strategy for Hallucination Reduction in MLLMs
[2] CHiP: Cross-modal Hierarchical Direct Preference Optimization for Multimodal LLMs
[3] Automated multi-level preference for mllms

---

> ### Author Response · Authors · 2025-11-28
> **Response to W1 (Generalization Claims) and Q1 (Learnable Weighting)**
>
> We thank the reviewer for the detailed review and constructive comments. Below are our clarifications to all the questions.
>
> We hope that the responses can address the reviewer’s concerns and we truly welcome further discussion.
> # **Weakness 1: Model Generalization (LLaVA-v1.5 and Qwen2.5-VL)**
>
> In the original paper, our primary experiments are conducted on the LLaVA-v1.5 series, but we also include experiments on Qwen2.5VL to demonstrate the cross-model generalization of our method.
>
> We would like to explain that **Figure 3 in the original PDF** (middle-right of page 7) already reports our results on the **Qwen2.5-VL**. On two widely used and comprehensive hallucination benchmarks, **MM-Hal** and **AMBER**, our Cat-PO achieves consistent and substantial gains over the base models and DPO method. The summarized results are as follows:
>
> | Model                 | MM-Hal Score (↑)| Hal-Rate (↓) | AMBER-F1 (↑) |
> |-----------------------|--------------|------------------|----------|
> | **Qwen2.5-VL-3B** (base)  | 2.89            | 43%                | 87.7        |
> | + DPO                 | 3.32            | 39%                | 90.1        |
> | + **Cat-PO (ours)**       | 3.51            | 32%                | 91.3        |
>
> ---
>
> # **Question 1: Learnable Fusion Weights** (For the three signals)
> We thank the reviewer for the insightful suggestion. Following your idea, we modified the fusion coefficients of the three signals to be learnable parameters: s_i = alpha * S_global + beta * S_local + (1 - alpha - beta) * S_semantic, with alpha and beta are the learnable parameters. Then, we jointly optimized alpha and beta with the model parameters within training loss (Essentially DPO loss). The summarized results are as follows:
>
> | Model                 | MM-Hal Score (↑) | Hal-Rate (↓) |
> |-----------------------|--------------|------------------|
> | Cat-PO (our general)  | 2.76            | 49%                |
> | Cat-PO with learnable fusion             | 2.55            | 50%                |
>
> The results show that introducing learnable coefficients yields an MM-Hal score of 2.55, below the original performance, with no observable reduction in hallucination. Upon further analysis, we believe this reveals two key points:
> (1) **Limitation of DPO loss:** DPO maximizes the likelihood difference between Chosen and Rejected responses rather than evaluating whether the learned weight allocation is intrinsically correct. Without direct supervision (ground-truth weights), the learnable coefficients tend to overfit statistical noise to minimize training loss.
> (2) **Stability and interpretable of the fixed uniform weighting:** The main benefit of Cat-PO comes from the complementary nature of the three cross-modal internal signals (global attention, entropy-based locality, and semantic similarity), rather than from a more complex fusion structure. The original fixed, uniform weighting appears to be a more stable and effective choice.
>
> Therefore, we plan to retain the fixed and uniform fusion design, which aligns with our goal of a simple and interpretable mechanism. We have added this ablation study and discussion to in the updated version of the paper. We again thank the reviewer for helping us strengthen the methodological justification.
>
> **Note:** These details can also be found in **Section 4.5 Analysis Experiment** (in page 9) of **the revised PDF (highlighted in yellow)**.

---

> ### Author Response · Authors · 2025-11-28
> **Response to Q2 (Comparative Evaluation), Part-1/2**
>
> # **Question 2: Comparisons with Related Works**
> We sincerely thank you for pointing out these three related works: **CHiP**, **TARS**, and **AMP**. Below, we provide structured comparisons between Cat-PO and these methods. For clarity, each comparison is summarized in a table.
>
> ## Comparison 1: CHiP (related work) vs. Cat-PO (ours)
>
> | Aspect                   | CHiP                                                                                                        | Cat-PO (ours)                                                                                                                          |
> |--------------------------|-------------------------------------------------------------------------------------------------------------|-----------------------------------------------------------------------------------------------------------------------------------------|
> | 1. Source of token-weights | Token-level weights are derived from **text-only probabilistic signals**, without image content, e.g., sequence-level KL on logits. | Token-level weights are **dynamically adaptive**, directly derived from internal **cross-modal attention** and **semantic similarity**, and thus reflect the true association between each token and the image. |
> | 2. Role of the weights   | Token-aware optimization is used as an **additional regularization term** on the loss, mainly to balance multi-granularity losses and prevent degeneration while preserving diversity. | Token-level weights are explicitly designed as **visual relevance rewards** for **fine-grained reweighting** of the DPO objective, enabling the model to correct hallucinations according to visual dependence. |
> | 3. Hierarchical nature   | The hierarchy (response-level / span-level / token-level) is entirely on the **text side**, without modeling the visual content. | The hierarchy is fully **cross-modal**: global attention, local patch entropy, and semantic similarity are all computed from deep **vision–language interactions** inside the model. |
>
> ---
>
> ## Comparison 2: TARS (related work) vs. Cat-PO (ours)
>
> | Aspect                                          | TARS                                                                                                                              | Cat-PO (ours)                                                                                                                                                           |
> |-------------------------------------------------|-----------------------------------------------------------------------------------------------------------------------------------|------------------------------------------------------------------------------------------------------------------------------------------------------------------------|
> | 1. Motivation / Design idea               | **Adversarial** perspective: focuses on bad / **visual-agnostic tokens**, and perturbs them (masking / replacement) to **destroy distracting signals** that cause hallucinations. | **Incentive-based** perspective: focuses on “good” or **visually grounded tokens**, and **rewards / upweights** them. The core idea is to **enhance informative signals**, letting the model prioritize visually relevant tokens. |
> | 2. Depth of visual relevance modeling           | Relies mainly on a **single metric**: cosine similarity between image features and token embeddings. | Proposes a **hierarchical** relevance measure: beyond cross-modal semantic similarity, we also exploits internal **cross-modal attention** (global) and **patch entropy** (local). |
> | 3. Optimization objective | Uses a **min–max adversarial** optimization procedure with additional frequency-domain regularization. | Designs a **KL-based customized loss** with token-level rewards that integrates naturally into DPO framework. |

---

> ### Author Response · Authors · 2025-11-28
> **Response to Q2 (Comparative Evaluation), Part-2/2**
>
> We sincerely thank you for pointing out these three related works: **CHiP**, **TARS**, and **AMP**. Below, we provide structured comparisons between Cat-PO and these methods. For clarity, each comparison is summarized in a table.
>
> **Note:** These details can also be found in **Appendix A.3** (in page 13) of **the revised PDF (highlighted in yellow)**.
>
> ---
>
> ## Comparison 3: AMP (related work) vs. Cat-PO  (ours)
>
> | Aspect                                     | AMP                                                                                                                                             | Cat-PO (ours)                                                                                                                                                                        |
> |--------------------------------------------|--------------------------------------------------------------------------------------------------------------------------------------------------|---------------------------------------------------------------------------------------------------------------------------------------------------------------------------------------|
> | 1. Granularity (response-level vs. token-level) | Operating at the **response-level**, it generalizes binary preference comparison (A > B) to multi-level ranking (A > B > C). | Operating at the **token-level**, Cat-PO assigns **token-level rewards**, explicitly distinguishing which tokens are more relevant to the visual input. |
> | 2. Source of data / signals                | Strongly depends on **external data engineering**: requires multiple MLLMs of different sizes or data scales for generation, plus an **auto-check mechanism** for data filtering. | Relies solely on the model’s **intrinsic multimodal capabilities**, without introducing external tools. It directly leverages internal **cross-modal attention** and **semantic similarity**.              |
> | 3. Optimization objective                   | Loss is essentially a **sum of multiple DPO losses** plus an additional penalty term on the best response to avoid probability collapse.         | Loss is **token-reward based**: beyond preference alignment, it performs **fine-grained attention correction**. Through global / local / semantic rewards, the model is explicitly encouraged to focus on visually grounded tokens and down-weight visually irrelevant ones during DPO training. |

---

### Author Response · Authors · 2025-12-03
**Summary of Rebuttal (Part 1/2)**

Dear AC, SAC, PC and Reviewers,

Thank you very much for your valuable contributions to our work. To assist the newly assigned AC and help reduce their workload, we provide below a summary of the key points from the reviews and our responses.

---

We respectfully invite you to consult our detailed rebuttal and the revised PDF, which include substantial new analyses and comprehensive responses to all reviewer comments. We believe that all concerns have now been thoroughly addressed.

---

### **Strength**
We are pleased that the reviewers recognized our work as having:
- **"Clear motivation"**, "well-specified method" and "clear writing" (k6vM, vajx),
- "**Fine-grained** token-level rewards are derived from MLLM internal signals, **without external dependencies**" (e8nW, k6vM, vajx),
- **"The experiments are sufficiently comprehensive"** and **"consistent empirical gains"** (k6vM, vajx).

---

### **Concerns and Our Addressing (Part 1/2)**
We systematically responded to all comments and strengthened the paper in several ways:

- **The Only Negative Rating** (score 4, by reviewer k6vM): ------***Reviewer k6vM: "Would Raise Score once Concerns are Resolved."***
   We have effectively resolved k6vM’s main concern that “*incorrect attention may lead to inaccurate rewards and amplify hallucinations*,” **which stemmed from an incomplete understanding of our method as relying solely on attention.**
   - **Our Clarification:** Cat-PO’s token-rewards is fused from 3 complementary signals: cross-modal attention (global), patch entropy (local), and cross-modal similarity (semantic). In practice, the raw global attention accounts for only about 25% of the token-rewards.
   - **Our Theoretical Analysis** (*in Rebuttal Text to k6vM, in Section 4.5 of Revised PDF* ): We provide a detailed analysis of the 3 signals’ roles, their organic fusion mechanism, the different treatment of chosen (y+) vs. rejected (y-) samples, and macro-level evidence from main and full ablation experiments. These analyses show that even in rare edge cases where attention is unreliable, the multi-signal design keeps the reward computation robust.
   - **Our Visualization Evidence** (*in Figure 7 of Revised PDF* ): The visualizations demonstrate that when attention is biased in edge cases, other signals (e.g., semantic similarity) act as a strong "safety barrier" that corrects reward deviations and prevents the model from incorrectly rewarding.
   - **Conclusion:** We think these results satisfy the reviewer’s stated condition (*reviewer k6vM’s original words were: **“ I would raise my score if the authors could address this issue, either through a theoretical discussion or via visualizations that demonstrate how their method remains robust in such scenarios.”***). We believe this robustness concern has been thoroughly resolved.

---

> ### Author Response · Authors · 2025-12-03
> **Summary of Rebuttal (Part 2/2)**
>
> ### **Concerns and Our Addressing (Part 2/2)**
> - **Other Addressing**
>    1. **Learnable fusion strategies and stability**  (*in Section 4.5, Line 447, highlighted in yellow;  in Table 4*)
>    As suggested by e8nW and vajx, we additionally explored a learnable fusion of the 3 signals for token-rewards. The experiments show that our original Cat-PO design offers superior stability and interpretability, while achieving competitive performance, thus justifying the chosen fixed fusion scheme.
>
>    2. **Training stability and gradient analysis** (*in Section 4.5, Line 417, highlighted in blue;  in Figure 6*)
>    Following vajx’s suggestion, we track the loss and reward margin throughout training. The newly added visualizations show that the loss decreases relatively smoothly, the reward margin increases steadily without significant oscillation, and the monitored gradient norms also remain stable.
>
>    3. **Evaluation under edge cases**  (*in Section 4.5, Line 442, highlighted in blue;  in Table 3*)
>    As suggested by vajx, we introduce experiments on highly biased and adversarial edge cases. The results indicate that our Cat-PO method remains relatively robust even in these challenging scenarios.
>
>    4. **Interpretability and high-frequency word rewards** (*in Rebuttal Text to k6vM: Response to Q1, Q2, Q3*)
>     Following k6vM’s suggestion, we add statistics showing that high-frequency function words (e.g., “the”, “a”) have clearly lower attention scores and rewards, supporting that Cat-PO does not blindly reward frequent function words.
>
>    5. **Comparison with related works** (*in Appendix A.3, Line 701, highlighted in yellow*)
>    As suggested by e8nW and vajx, we systematically compare Cat-PO with related methods including TARS, CHiP, TDPO, TG-DPO, and AMP, Clarifying the distinctiveness and novelty of Cat-PO.
>
>    6. **Clarifications on content that some reviewers may overlooked**
>       - **Model Generalization** : Cat-PO has already been shown to generalize across different model sizes (LLaVA-v1.5-7B and -13B) and different architectures (LLaVA vs. Qwen2.5-VL).    (*in Section 4.3, Line 343; Table 1 and Figure 3*)
>       - **Training Overhead** : We compare Cat-PO with DPO in terms of training time and memory, showing that Cat-PO introduces only moderate extra time cost while keeping the peak memory almost unchanged.   (*in Section 4.5, Line 466; Table 5*)
>
> ---
> ### **Summary**
> **The current review status is that two of the three reviewers initially gave positive scores of 6** and expressed supportive attitudes. The only reviewer who gave a score of 4 raised a core concern about possible attention misalignment, which stemmed from an incomplete understanding of our method as relying solely on attention.
>
> We believe that **all concerns have now been thoroughly addressed**, which also **aligns with reviewer k6vM’s stated condition for raising the score** (original score: 4). In addition, given our detailed and thorough responses, we also believe that the other two reviewers (original score: 6) may further update their assessments.
>
> Considering the **relatively high initial average score, the overall positive evaluation** of our motivation, method, and experiments, and the **substantial new theoretical analyses, statistical and experimental results, and visual evidence added during the rebuttal**, we sincerely hope our work will receive your positive consideration.
>
> ---
>
> Above, we have provided a concise summary of all reviewer comments and our corresponding responses, and we hope that this overview will be helpful for the AC’s evaluation.
>
> **We sincerely thank the reviewers, AC, SAC, and PC for the time and effort devoted to assessing our submission.** Your constructive feedback has substantially improved the clarity and quality of the paper.
>
> Authors

---

### Meta-Review · Area_Chair_6DXu · 2026-01-09

**Summary:**

This paper introduces Cat-PO (Cross-modal Adaptive Token-rewarded Preference Optimization) to reduce hallucinations in Multimodal Large Language Models. Unlike standard DPO, Cat-PO uses the model’s internal signals—cross-modal attention and semantic similarity—to calculate token-level rewards based on global, local, and semantic visual relevance.
By rewarding tokens with strong visual grounding and penalizing ungrounded ones, Cat-PO achieves finer alignment between text and images without external tools. Testing on benchmarks like AMBER and MM-Hal shows that Cat-PO significantly improves factuality and reduces hallucinations in models like LLaVA and Qwen with minimal computational overhead.

Every reviewer praised the shift from aggregate rewards to token-level granularity. By assigning rewards based on global, local, and semantic relevance, Cat-PO offers a more precise mechanism for hallucination suppression than standard DPO. Reviewers also highlighted that the method is "self-contained." Because it uses internal signals (attention, entropy, similarity) rather than external APIs or detectors, it maintains a low computational overhead (+0.07% peak memory). Multiple reviewers noted the consistent gains across benchmarks like AMBER and MM-Hal, confirming that the theoretical approach translates into measurable improvements in factuality.

**Reviewer Concerns:**

1. The most significant conceptual concern is the risk of “circular reasoning" by reviewer k6vM.  If a model is already hallucinating (e.g., attending to a "dog" that isn't there), its internal attention signals will be "highly relevant" to that hallucination. Reviewer 2 points out that Cat-PO might reinforce incorrect attention patterns rather than correcting them.

The authors provided additional explanation and visualization to illustrate the motivation and the robustness of the three way fusion by combining attention and global signals. It is not fully convincing but indeed provide justification to demonstrate the robustness of the approach.


2. Reviewer vajx also noted that the fusion of the three relevance scores relies on fixed coefficients and tanh-smoothing without a strong theoretical or learned basis, describing the integration as somewhat heuristic.

The authors listed the principles they designed the function, which confirms reviewer's comment the choice is somewhat heuristic. One option is to list ablation experiment numbers to justify the choice.

**Reviewer Scores:**

Reviewer e8nW and vajx are likely to remain the positive score.

Reviewer k6vM commented that he will raise the score if his concern of circular reasoning is addressed. The authors indeed answered the question through visualization as requested. So it is likely reviewer k6vM will turn positive to this paper.

---

### Decision · Program_Chairs · 2026-01-26

Accept (Poster)